# Theoretical Study on the Electronic Structure and Magnetic Properties Regulation of Janus Structure of M’MCO_2_ 2D MXenes

**DOI:** 10.3390/nano12030556

**Published:** 2022-02-06

**Authors:** Panpan Gao, Minhui Song, Xiaoxu Wang, Qing Liu, Shizhen He, Ye Su, Ping Qian

**Affiliations:** 1Beijing Advanced Innovation Center for Materials Genome Engineering, University of Science and Technology Beijing, Beijing 100083, China; gaopanpan@ustb.edu.cn (P.G.); mhuisong@163.com (M.S.); wangxx@dp.tech (X.W.); liuqing6903@163.com (Q.L.); hsz0430@163.com (S.H.); 2School of Mathematics and Physics, University of Science and Technology Beijing, Beijing 100083, China; 3Department of Physics, National University of Singapore, Singapore 117551, Singapore

**Keywords:** janus, MXenes, magnetic properties, DFT

## Abstract

Motivated by the recent successful synthesis of Janus monolayer of transition metal (TM) dichalcogenides, MXenes with Janus structures are worthy of further study, concerning its electronic structure and magnetic properties. Here, we study the effect of different transition metal atoms on the structure stability and magnetic and electronic properties of M’MCO_2_ (M’ and M = V, Cr and Mn). The result shows the output magnetic moment is contributed mainly by the d orbitals of the V, Cr, and Mn atoms. The total magnetic moments of ferromagnetic (FM) configuration and antiferromagnetic (AFM) configuration are affected by coupling types. FM has a large magnetic moment output, while the total magnetic moments of AFM2’s (intralayer AFM/interlayer FM) configuration and AFM3’s (interlayer AFM/intralayer AFM) configuration are close to 0. The band gap widths of VCrCO_2_, VMnCO_2_, CrMnCO_2_, V_2_CO_2_, and Cr_2_CO_2_ are no more than 0.02 eV, showing metallic properties, while Mn_2_CO_2_ is a semiconductor with a 0.7071 eV band gap width. Janus MXenes can regulate the size of band gap, magnetic ground state, and output net magnetic moment. This work achieves the control of the magnetic properties of the available 2D materials, and provides theoretical guidance for the extensive design of novel Janus MXene materials.

## 1. Introduction

Over the past decade, the two-dimensional (2D) materials have received significant interest since the discovery of graphene [1,2,3]. Compared with bulk materials, more atoms on the surface of 2D materials are exposed, which is caused by reduced dimensionality. This improves the utilization rates of atoms and makes regulation of band structure and electronic properties easier, thus enabling MXenes to exhibit novel physical and chemical properties [4,5,6]. Recently, a new family of 2D transition metal (TM) carbides and nitrides, MXenes, has received more and more attention [7,8]. MXenes have the general formula M_n+1_X_n_T_x_, where M stands for early TM, X represents C or N, T_x_ indicates the surface functional groups O, OH, or F, and n = 1, 2, or 3 [9,10]. Usually, MXenes are synthesized by selective etching A layers (A is an element from the A-group 13 or 14) in the MAX phase, using hydrofluoric acid (HF) solutions [11,12,13]. Sue to their excellent electrical [14], optical [15,16,17,18], and mechanical properties [19], MXenes have been widely applied in electronic devices [20], catalysis [21,22,23,24], magnetic storage [25], energy conversion, and storage systems [26,27,28]. Thus far, more than 30 kinds of MXenes have been synthesized in experiments, and more kinds of materials have been theoretically predicted [29,30].

The electronic and optical properties of MXenes with symmetrical configuration have been extensively studied. Previous theoretical investigations have shown that, without surface functionalization, Cr_2_C is half metallic and ferromagnetic (FM) configuration [31], V_2_C exhibits metallic and antiferromagnetic (AFM) configuration [32], and Ti_n+1_C_n_ and Ti_n+1_N_n_ (n = 1–9) show magnetic configuration [33]. While functionalized MXenes alter magnetism, others, such as Cr_2_CX_2_ (X=OH, O and F), V_2_CX_2_ (X=F, OH), and Ti_2_CO_2_, are semiconductors [31,32,33,34,35]. Structural symmetry is a key factor in determining the electronic properties of 2D materials [36,37]. If structural symmetry is broken, it is desirable for 2D materials to have electronic and magnetic properties.

Inspired by the successful synthesis of Janus monolayers of TM dichalcogenides [38], MXenes with Janus structures are worth studying further, especially concerning their electronic structures and magnetic properties. Janus refers to MXenes that break the symmetry through asymmetric surface functional groups or different types of TM elements [39]. A previous report has theoretically studied the electronic and magnetic properties of Janus MXenes; it indicated that, by selecting an appropriate terminal group of upper and lower surfaces, the band gap of Janus MXenes can be successfully adjusted to different regions [40]. Therefore, we consider that different TM atoms may also regulate the charge and chemical environment around the atom, which causes Janus MXenes to exhibit significantly different electronic and magnetic properties.

In this paper, using first-principles calculations, we employed M’MCO_2_ (M’ and M stands for V, Cr, and Mn) configurations to investigate the effect of different types of TM on the structure and the magnetic and electronic properties under the same functional groups. We constructed different magnetic configurations (nonmagnetic (NM), FM, and AFM) for each M’MCO_2_ structure, researched their magnetic properties, and screened out the magnetic ground state. Then, we studied the electronic structure of the magnetic ground state. The results showed that Janus MXenes can adjust the band gap, the magnetic ground state, and the net output magnetic moments, which is a very good control method. Due to its asymmetric structure, Janus MXenes can flexibly control the magnetism of a system by applying small electric fields. This work provides theoretical guidance for the realization of the magnetic controllability of MXene materials.

## 2. Materials and Methods

All calculations were carried out using the Vienna ab initio simulation package (VASP), based on density functional theory (DFT) [41,42]. The generalized gradient approximation (GGA), with the Perdew–Burke–Ernzerhof (PBE) functional, was used for the exchange and correlation functional [43,44]. Interactions between electrons and nuclei were described by the projector augmented wave (PAW) method [45]. A plane wave kinetic energy cutoff 600 eV was employed. The convergence criteria of total energy and atomic force for each atom were set to 10^−5^ eV per unit cell and 10^−4^ eV/Å, respectively.

To account for the energy of localized 3D orbitals of TM atoms properly, the Hubbard “U” correction was employed within the rotationally invariant DFT + U approach proposed [46]. The spin-polarized DFT + U correction [47,48] was applied to strongly correlated Cr, V, and Mn atoms with the typical U = 4 eV value. The specific U value does not change the predicted magnetic ordering nor the easy axis determination [49,50]. The cutoff kinetic energy for plane waves was set to 600 eV. Considering the van der Waals interaction between layers, the Becke–Jonson attenuation DFT + D3 method was performed for empirical correction.

A vacuum spacing of 20 Å along the M’MCO_2_ normal was used to avoid the interactions caused by the periodic boundary condition. The Brillouin zone (BZ) was sampled using 11 × 11 × 1 Γ-centered, k-point Monkhorst-Pack grids for the calculations of relaxation and electronic structures for NM, FM, and AFM1 primitive cells. Additionally, k-mesh was decreased to 6 × 12 × 1 for 2 × 1 × 1 AFM2 and AFM3 supercells. In the static self-consistent calculation, k-point grid sampling of 20 × 20 × 1 was used for the primitive cell, and k-point grid sampling of 12 × 24 × 1 was used for the 2 × 1 × 1 supercell.

## 3. Results

### 3.1. Stable Structures of Janus MXenes

The monolayer M_2_C MXene is a centered honeycomb structure with P3m1 symmetry, in which the 2D hexagonal C atom is sandwiched between two hexagonal M atoms. There are four possible configurations for O atoms absorbed on the M atom [51]: (a) O atoms located right above the M atoms (top sites); (b) O atoms located at the hollow sites of adjacent C atoms (hcp sites); (c) O atoms located at the hollow sites of contralateral M atoms (fcc sites); (d) on the one side, O atoms are at fcc sites, and on the other side, O atoms are at the hcp sites. According to previous research by Tan [52] and Wang [53], (b) configuration is stable for CrMnCO_2_ and Cr_2_CO_2_ and (c) configuration is stable for VCrCO_2_, VMnCO_2_, V_2_CO_2_, and Mn_2_CO_2_. So, we selected those configurations for the following calculations. Figure 1 shows the structures of symmetric V_2_CO_2_, Cr_2_CO_2_, and Mn_2_CO_2_. The arrangement of atoms observed from the top and bottom is the same. Figure 2 shows the structures of Janus MXenes VCrCO_2_, VMnCO_2_, and CrMnCO_2_. It can be seen that the arrangement of atoms seen from the top and bottom is different, so the symmetry of VCrCO_2_, VMnCO_2_, and CrMnCO_2_ is lower than that of V_2_CO_2_, Cr_2_CO_2_, and Mn_2_CO_2_, which is consistent with the symmetry of their space group. The basic information of their lattice parameters is shown in Table 1.

Many compounds of V, Cr, and Mn are magnetic [39]. We calculated the total energy of the non-spin-polarized system and the spin-polarized system, respectively, by using standard DFT method: the result is shown in Table 2. It can be seen that, except V_2_CO_2_, the total energy of the spin polarization is lower than that of the non-spin polarization. When taking spin polarization into account, obvious magnetic moment can be observed in magnetic atoms. Therefore, the ground states of VCrCO_2_, VMnCO_2_, CrMnCO_2_, Cr_2_CO_2_, and Mn_2_CO_2_ must be magnetic, while the ground state of V_2_CO_2_ is NM. Although the ground states of V_2_CO_2_ are NM, the difference between the NM and the magnetic state is very small (about 0.0002 eV). When considering the spin polarization in the system, the V atom has about 1 μB/atom sized magnetic moment. Under certain conditions, NM may become magnetic, so it is necessary to study its magnetic properties.

Considering the magnetism of the TM atoms, we employed the FM and AFM order for each M’MCO_2_ configuration, as shown in Figure 3. For FM configurations, the magnetic moments of M atoms are parallel, while for AFM configurations, the magnetic moments are antiparallel with each other. According to the different coupling kinds between atoms, we constructed three different AFM configurations, named AFM1, AFM2, and AFM3. AFM1 configuration is characterized by intralayer FM coupling and interlayer AFM coupling; AFM2 configuration is characterized by intralayer AFM coupling and interlayer FM coupling; AFM3 configuration is characterized by AFM coupling of atoms both intralayer and interlayer. The initial models of NM, FM, and AFM configurations are completely the same, but due to the different coupling models of magnetic atoms, after structure optimization, different configurations will have different lattice parameters, total energy, electronic structure, and other aspects. Compared to the initial models, the lattice constants of NM hardly change, while the corresponding FM and AFM are slight increased. Meanwhile, FM and AFM configurations symmetries are also reduced, which is related to the coupling between magnetic atoms after spin polarization, as shown in Appendix A.

### 3.2. Magnetic Properties

The spin polarization can be corrected by adopting the DFT + U method, but it will introduce additional situ coulomb interaction energy. However, U is not added in the case of the non-spin-polarized system, which means that the total energy of the two system do not have comparability. Therefore, in the following analysis, we study the electronic structures and magnetic properties of the ground states of VCrCO_2_, VMnCO_2_, CrMnCO_2_, V_2_CO_2_, Cr_2_CO_2_, and Mn_2_CO_2_ configurations under spin polarization, regardless of NM. The calculated total energy is shown in Table 3. The coupling effect between magnetic configurations is different, making the corresponding energy and other properties different. The total energy difference between different magnetic states is very small, which means that the ground states of AFM and FM are unstable in a specific environment. For each M’MCO_2_ configuration, the configuration with the lowest energy is its most stable configuration state—the magnetic ground state; moreover, this is the focus of the present analysis and study. From the value, we can obtain that the magnetic ground states of VCrCO_2_, VMnCO_2_, and Cr_2_CO_2_ are FM, and that the ground state energies are −37.321 eV/u.c, −36.936 eV/u.c, and −35.488 eV/u.c, respectively. The magnetic ground state of CrMnCO_2_ is AFM3, and the ground state energy is −35.707 eV/u.c. The ground state of V_2_CO_2_ is NM, the magnetic ground state is AFM1, and the magnetic ground state energy is −38.020 eV/u.c. The magnetic ground state of Mn_2_CO_2_ is AFM2, and the ground state energy is −35.989 eV/u.c. In previous studies on the magnetic properties of MXenes, Mohammad Khazaei [35] and Tan [54] calculated the magnetic ground state of Cr_2_CO_2_ as FM, while Hu [32] obtained the magnetic ground state of V_2_CX_2_ (X=F, OH) as AFM. The conclusion of these studies is consistent with our calculated results. Additionally, we can conclude that, when replacing TM atoms, symmetric MXenes V_2_CO_2_, Cr_2_CO_2_, and Mn_2_CO_2_ become Janus MXenes VCrCO_2_, VMnCO_2_, and CrMnCO_2_, respectively; the magnetic ground states will change, and Janus MXenes can regulate the magnetic ground states.

The magnetic moments of all magnetic configurations of each M’MCO_2_ are summarized in Appendix A. Additionally, we drew the curves of the magnetic moments under different magnetic configurations and M’MCO_2_ configurations, as shown in Figure 4. From the contribution of atomic species to the magnetic moments, we can find that the magnetic moments of C and O are close to 0 and the magnetic moment is mainly contributed by the TM atoms V, Cr, and Mn. In addition, the curves satisfied μ(Mn) > μ(Cr) > μ(V) because of the different electron numbers—Mn has one more electron than Cr, and Cr has one more electron than V. In the FM configurations, the order of total magnetic moment is μ(Mn_2_CO_2_) > μ(MnCrCO_2_) > μ(Cr_2_CO_2_) > μ(VMnCO_2_) μ(VCrCO_2_) > μ(V_2_CO_2_). The large the specific gravity of Mn in the configuration, the greater the net magnetic moment; the large the specific gravity of V, the smaller the net magnetic moment, which agrees with μ(Mn) > μ(Cr) > μ(V). Because the magnetic moment of V atom is small, when the TM atom in MXene is V, the stable configuration tends to be nonmagnetic. This means that the ground state of V_2_CO_2_ is nonmagnetic, while the ground states of VCrCO_2_, VMnCO_2_, CrMnCO_2_, Cr_2_CO_2_, and Mn_2_CO_2_ are magnetic. As the magnetic order changes, the magnetic moments of V, Cr, and Mn have little change, and it basically maintains a horizontal trend within the range of 0.50 μB. It indicates that the type of TM atom is a decisive factor for the magnetic moment; moreover, the environment of atoms and the coupling mode between atoms have little effect on the magnetic moment.

Meanwhile, the total net magnetic moment is greatly affected by the magnetic configuration. It is clear that FM shows obvious magnetic moment, in which the magnetic moment of magnetic atoms is in the same direction. The total magnetic moment is similar to the algebraic sum of magnetic moments of magnetic atoms, so it has a large magnetic moment output. As for AFM1, Janus MXenes show obvious magnetic moment, while the net magnetic moment of symmetric MXenes is almost 0. Given that the intralayer atoms are composed of FM coupling, the different magnetic moments of the top and bottom atoms cannot completely cancel the Janus MXene; therefore, it has net magnetic moments and exhibits ferromagnetism. In contrast to the symmetric MXenes with the same top and bottom atoms, the magnetic moment can be completely cancelled out, so the net magnetic moment is 0 and it exhibits anti-ferromagnetism. The net magnetic moments of AFM2 and AFM3 are almost 0, a value which does not show magnetic moment externally. The reason is that both the intralayer coupling is AFM and the adjacent electrons have opposite spin directions, which causes the net magnetic moment of top and bottom layers to be 0, a value which does not show magnetic moment externally. We discover that the higher the symmetry of the magnetic moment, the lower the total energy, and the more stable this configuration will be. Besides, there is modulation between different TM atoms in Janus MXenes. When V atoms are replaced with Cr in V_2_CO_2_, the magnetic moment of the V atoms becomes smaller; meanwhile, when Mn atoms are replaced with Cr in Mn_2_CO_2_, the magnetic moment of the Mn atoms becomes large—the modulation effects of Cr atoms on V_2_CO_2_ and Mn_2_CO_2_ are different. As for the Cr_2_CO_2_ configuration, replacing the Cr atoms with V or Mn all will cause the magnetic moment of the Cr atoms to become large. Therefore, we conclude that Janus MXenes can manipulate the size of magnetic moment.

### 3.3. Electronic Properties

For the above magnetic ground states, we further explored their electronic properties, and calculated their band structures and densities of state (DOS), respectively. Figure 5 shows the band structures of each M’MCO_2_. Additionally, we found that the spin-up and spin-down curves of Mn_2_CO_2_, CrMnCO_2_, and V_2_CO_2_ almost completely coincide, which is consistent with AFM configurations; while the spin-up and spin-down curves of VCrCO_2_, VMnCO_2_, and Cr_2_CO_2_ split, which are almost the only spin-up curves near the Fermi level—this finding is consistent with FM configurations. Therefore, we found that the configuration of Mn_2_CO_2_, CrMnCO_2_, and V_2_CO_2_ are FM, and the configuration of VCrCO_2_, VMnCO_2_, and Cr_2_CO_2_ are FM.

As for M’MCO_2_ with AFM configurations, near the Fermi level, the valence band and the conduction band of Mn_2_CO_2_ are clearly separated, and no band curve crosses the Fermi level. Moreover, the top valence band is near the point M (0.622), and the bottom conduction band is near the point K (1.358), illustrating that Mn_2_CO_2_ is an indirect band gap semiconductor, a finding that is in keeping with the results of Zhou [55]. Meanwhile, both CrMnCO_2_ and V_2_CO_2_ have band curves crossing the Fermi level, and their top valence band and bottom conduction band are located at the point Γ, with a small band gap width, showing metallic character. With regard to M’MCO_2_, with FM configurations, it can be seen from Figure 5a,b,e that the red curves (spin-down) are distributed on both sides of the Fermi level, with large band gap widths, while the blue curves (spin-up) are densely distributed, crossing the Fermi level. The electronic states with spin-up make the band gap narrow—these are metallic materials. On the whole, we concluded that, when M is replaced with M’—causing non-Janus MXene to become Janus MXene—the band gap width was greatly reduced and the conductivity became better; therefore, Janus MXenes can regulate the band gap.

To facilitate the description of the modulation action of Janus MXenes, we plotted the density of state (DOS) of the magnetic ground state of M’MCO_2_, as shown in Figure 6. There are a large number of electrons near the Fermi level for VCrCO_2_, VMnCO_2_, CrMnCO_2_, V_2_CO_2_, and Cr_2_CO_2_, indicating that they have metallic properties. However, there are almost no electrons at the Fermi level for Mn_2_CO_2_, which can be considered as the forbidden band, and the band width is 0.7071 eV, indicating that it is a semiconductor. In addition, we found that the PDOS of the d orbitals of V, Cr, and Mn contributed most of the TDOS. For symmetric MXenes, the PDOS of d orbitals of V, Cr, and Mn are almost consistent with the TDOS of V_2_CO_2_, Cr_2_CO_2_, and Mn_2_CO_2_ in the energy range of −1~4 eV, −3~6 eV, and 2~6 eV, respectively. Meanwhile, for Janus MXenes, the TDOS are the synergy of d orbitals of different TM atoms. The d orbitals of the V atoms contributed most of the electrons in the following energy ranges: 0~2 eV for VCrCO_2_; −1~2 eV and 4~6 eV for VMnCO_2_. The d orbitals of the Cr atoms contributed most of the electrons in the following energy ranges: −2~0 eV and 2~6 eV for VCrCO_2_, −3~4 eV for CrMnCO_2_. The d orbitals of the Mn atoms contributed most of the electrons in the following energy ranges: −6~-1 eV and 2~4 eV for VMnCO_2_, and −5~−3 eV and 4~6 eV for CrMnCO_2_. The sum of the d orbitals of V, Cr, and Mn is almost equal to the TDOS, showing that the magnetism of the system is mainly derived from the d orbital electrons of magnetic atoms.

Since the magnetic moment of the system is proportional to the area integral of the upper and lower curves, the greater the difference between the spin-up and spin-down, the greater the split, and the greater the net magnetic moment. The upper curves and lower curves of V_2_CO_2_ and Mn_2_CO_2_ are nearly symmetric and the net magnetic moment is almost 0, which is consistent with the magnetic ground state of AFM. The magnetic ground state of Cr_2_CO_2_ is FM, and its upper and lower curves have a certain split, where the peak position energy of the upper curve is lower, and the peak position energy of the lower curve is higher. The area integral of DOS of the upper curve and the lower curve is close, within the energy range of −7~−3 eV, the spin-up electron state occupies the dominant position in the range of −3~2 eV, and the spin-down electron state occupies the dominant position in the range of 2~6 eV, which causes the FM configuration to have 4.184 μB/atom net magnetic moment.

When a V atom in V_2_CO_2_ is replaced by Cr or Mn atoms—becoming Janus MXene VCrCO_2_ or VMnCO_2_, with the magnetic ground state becoming FM—the upper and lower curves are clearly split. The main electron state is spin-up in the range of −2~2 eV and spin-down in the range of 2~6 eV. Below −2 eV, the peak of spin-up moves to low energy and the peak of spin-down moves to high energy. When a Cr atom is replaced with an Mn atom in Cr_2_CO_2_—becoming Janus MXene CrMnCO_2_, with the magnetic ground state becoming AFM3—the upper and lower curves are symmetric and the net magnetic moment is 0. In conclusion, we can judge the magnetic configuration of a system by the magnetic moment of the atoms, the energy band, and the DOS. Janus MXenes can regulate the band gap width, the magnetic ground state, and the net magnetic moment; this is a great adjusting method.

## 4. Conclusions

We investigated the magnetic properties and electron structures of M’MCO_2_ with different magnetic configurations. The following results were found: the magnetic ground states of VCrCO_2_, VMnCO_2_, and Cr_2_CO_2_ are FM; the magnetic ground state of CrMnCO_2_ is AFM2; the magnetic ground state of Mn_2_CO_2_ is AFM2; the magnetic ground state of V_2_CO_2_ is AFM1 and its ground state is NM. The band gap widths of VCrCO_2_, VMnCO_2_, CrMnCO_2_, V_2_CO_2_, and Cr_2_CO_2_ are no more than 0.02 eV, showing metallic properties; meanwhile, Mn_2_CO_2_ is a semiconductor, with a 0.7071 eV band gap width. Moreover, we determined the magnetic configuration of the systems through the magnetic moment, the energy band, and the DOS. Further analysis showed that Janus MXenes can adjust the band gap, magnetic ground state, and output net magnetic moment, resulting in smaller band gap widths and better electrical conductivities when compared with corresponding materials. These theoretical results provide guidance for further experimental verification and electronic device application.

## Figures and Tables

**Figure 1 nanomaterials-12-00556-f001:**
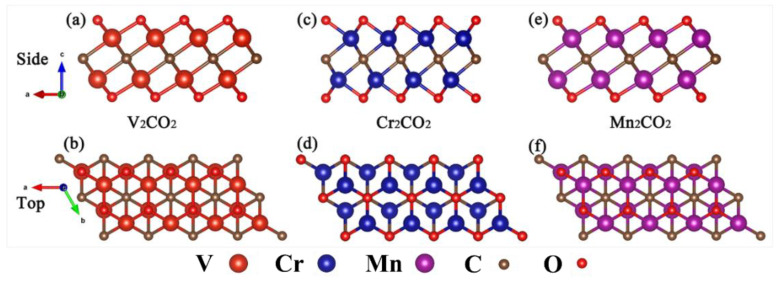
V_2_CO_2_, Cr_2_CO_2_, and Mn_2_CO_2_ structures: (**a**,**c**,**e**) and (**b**,**d**,**f**) are the side and top views of V_2_CO_2_, Cr_2_CO_2_, and Mn_2_CO_2_ structures, respectively.

**Figure 2 nanomaterials-12-00556-f002:**
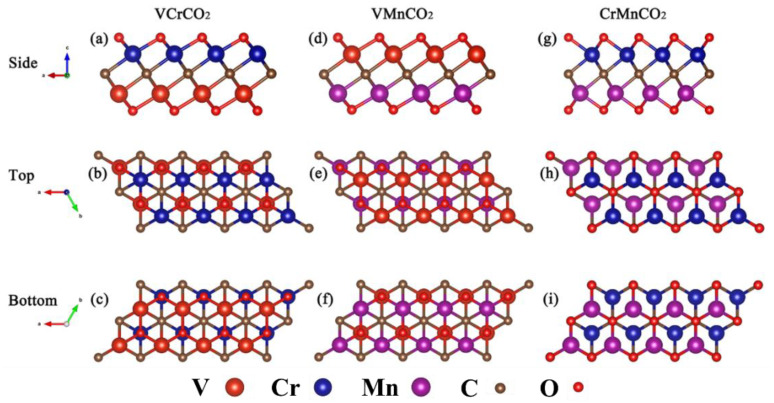
VCrCO_2_, VMnCO_2_, and CrMnCO_2_ structures: (**a**,**d**,**g**) and (**b**,**e**,**h**) and (**c**,**f**,**i**) are the side, top, and bottom views of VCrCO_2_, VMnCO_2_, and CrMnCO_2_ structures, respectively.

**Figure 3 nanomaterials-12-00556-f003:**
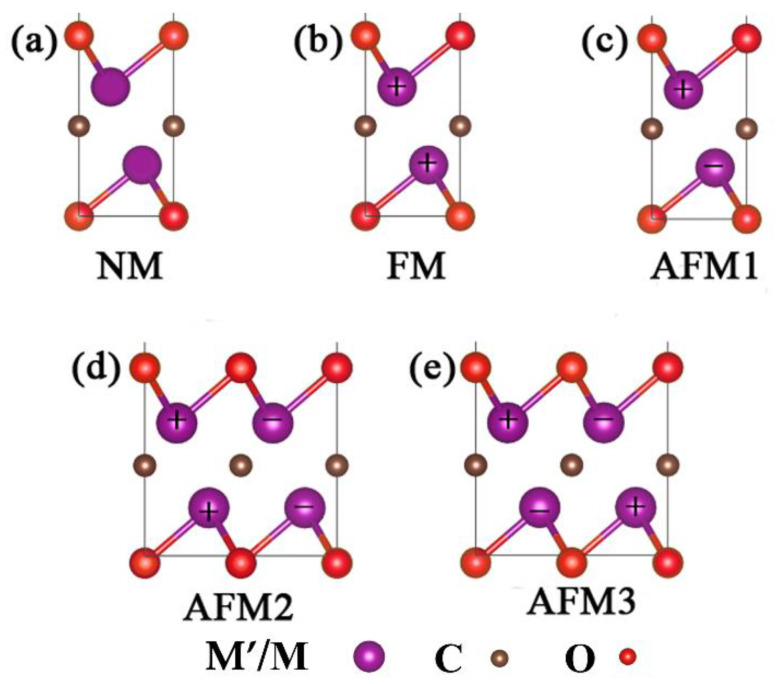
M’MCO_2_ magnetic state configurations: (**a**–**e**) represent NM, FM, AFM1, AFM2, and AFM3 configurations, respectively. In M’/M atoms, “+” represents spin-up and “−“ represents spin-down.

**Figure 4 nanomaterials-12-00556-f004:**
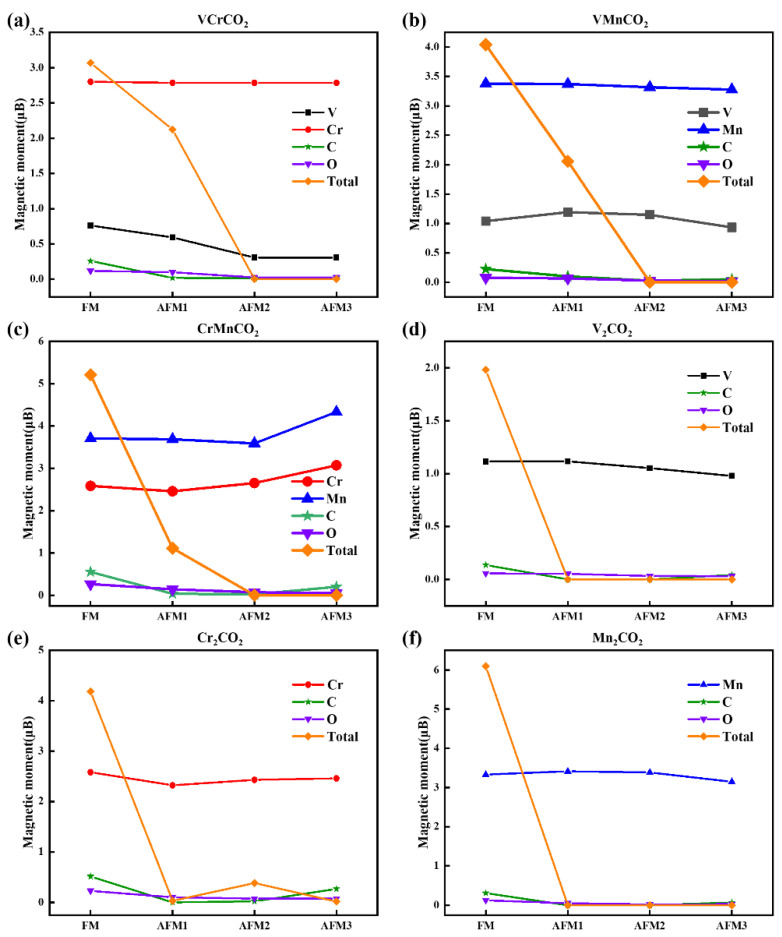
Magnetic moment curves under different M’MCO_2_ configurations. (**a**) VCrCO_2_ configuration; (**b**) VMnCO_2_ configuration; (**c**) CrMnCO_2_ configuration; (**d**) V_2_CO_2_ configuration; (**e**) Cr_2_CO_2_ configuration; (**f**) Mn_2_CO_2_ configuration.

**Figure 5 nanomaterials-12-00556-f005:**
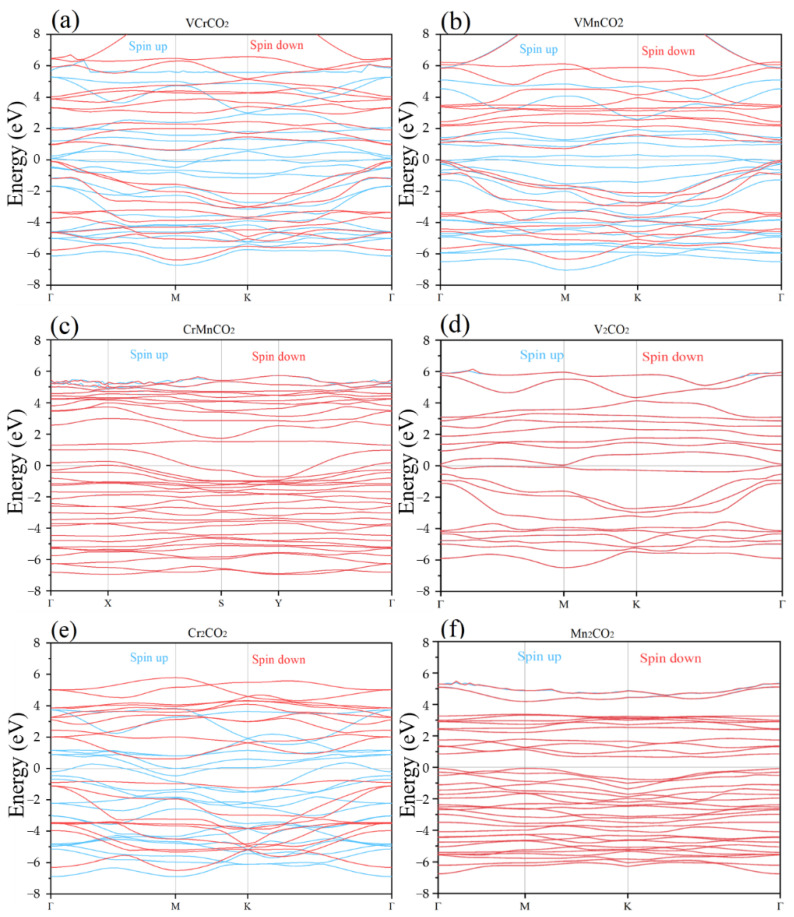
The band gap of M’MCO_2_. (**a**) VCrCO_2_; (**b**)VMnCO_2_; (**c**) CrMnCO_2_; (**d**)V_2_CO_2_; (**e**) Cr_2_CO_2_; (**f**) Mn_2_CO_2_.

**Figure 6 nanomaterials-12-00556-f006:**
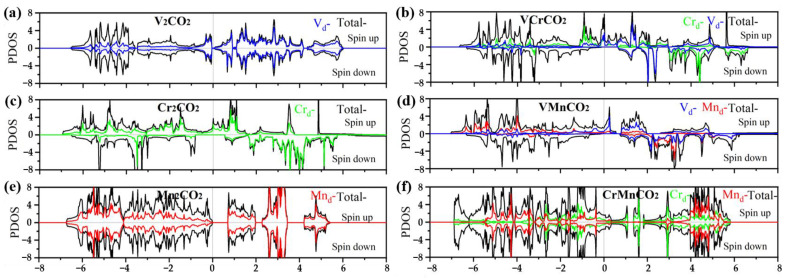
DOS of M’MCO_2_. (**a**) V_2_CO_2_; (**b**) VCrCO_2_; (**c**) Cr_2_CO_2_; (**d**) VMnCO_2_; (**e**) Mn_2_CO_2_; (**f**) CrMnCO_2_.

**Table 1 nanomaterials-12-00556-t001:** The lattice parameters of M’MCO_2_.

	VCrCO_2_	VMnCO_2_	CrMnCO_2_	V_2_CO_2_	Cr_2_CO_2_	Mn_2_CO_2_
Symmetry Group	P3m1 (C_3V-1_)	P3m1 (C_3V-1_)	P3m1 (C_3V-1_)	P$\bar{3}$m1 (D_3d-3_)	P$\bar{3}$m1 (D_3d-3_)	P$\bar{3}$m1 (D_3d-3_)
a/Å	2.88	2.89	2.66	2.88	2.68	2.87
b/Å	2.88	2.89	2.66	2.88	2.68	2.87
c/Å	21.81	21.81	21.81	21.81	21.81	21.81
α	90°	90°	90°	90°	90°	90°
β	90°	90°	90°	90°	90°	90°
γ	120°	120°	120°	120°	120°	120°

**Table 2 nanomaterials-12-00556-t002:** Total energy of spin-polarized and non-spin-polarized systems. NM stands for non-spin-polarized system, magnetic stands for spin-polarized systems, the unit of total energy is eV/u.c (unit cell).

	VCrCO_2_	VMnCO_2_	CrMnCO_2_	V_2_CO_2_	Cr_2_CO_2_	Mn_2_CO_2_
NM	−44.322	−42.926	−42.488	−45.597 (5)	−43.804	−40.701
Magnetic	−44.502	−43.451	−42.499	−45.597 (3)	−43.882	−41.431

**Table 3 nanomaterials-12-00556-t003:** The total energy of FM and AFM structures of M’MCO_2_, the unit of total energy is eV/u.c (unit cell).

	FM	AFM1	AFM2	AFM3
VCrCO_2_	−37.321	−37.215	−37.223	−37.223
VMnCO_2_	−36.936	−36.921	−36.921	−36.886
CrMnCO_2_	−34.778	−34.349	−34.743	−35.707
V_2_CO_2_	−38.013	−38.020	−37.911	−37.918
Cr_2_CO_2_	−35.488	−34.794	−35.263	−35.365
Mn_2_CO_2_	−35.934	−35.913	−35.989	−35.746

## Data Availability

The datasets generated during and/or analyzed during the current study are available from the corresponding author.

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
