# Peer review of "Theoretical Study on the Electronic Structure and Magnetic Properties Regulation of Janus Structure of M’MCO_2_ 2D MXenes"

_nanomaterials, 2022, doi:10.3390/nano12030556_

Round 1
Reviewer 1 Report
From the scientific point of view, the presented manuscript contains good theoretical research. Introduction has to be rewriten. Tenses, punctuation, singular/plurar forms and many others have to be thoroughly checked across whole manuscript.
Author Response
Thank you for your suggestions. I have revised the manuscript as you requested.

Reviewer 2 Report
Manuscript ID: nanomaterials-1569494
Title: Theoretical Study on Electronic Structure and Magnetic Properties Regulation of Janus Structure of M'MCO2 2D MXenes
Decision: Minor revision
Comment
The article reports the theoretical insights on the electronic and magnetic merits of the two-dimensional Janus Structure of M'MCO2 MXenes (M= V, Cr, and Mn). This includes the calculations of various magnetic configurations like nonmagnetic (NM), ferromagnetic (FM,) and antiferromagnetic (AFM)) for each M'MCO2 structure as well as the electronic structure as a function of MXenes composition. The result warranted that the Janus structure of MXenes can tailor the adjust the bandgap, magnetic ground state, and output net magnetic moments, originating from the asymmetric structure and Janus MXenes composition. The article is well written and supported with all needed calculations and is suitable for publication in Nanomaterials after addressing the following comments
- The figures numbers should be revised; Figure 4 is missing
- The authors should improve the resolution of the whole figures in the manuscript especially Figure 3-Figure 7
- In section 2 ‘’Materials and Methods’’ the authors claimed that ‘’ A vacuum spacing of 20 Å along the M'MCO2 normal was used to avoid ‘’ This spacing is very large compared to that usually observed in the experimentally prepared MXenes the authors should explain.
- Also, the authors should explain why he selected this formula M'MCO2 MXenes usually, Mxenes always have surface terminations of F, OH, and O
- The authors should add some values and numbers into the abstract
- The authors should compare the obtained results with elsewhere reports
- The similarity rate is around 23%, according to the iThenticate, so the authors should decrease it to be lower than 20 %
- In the introduction section, the authors should extend the introductory section about 2D materials and emphasize their advances over other materials, and can cite more references. e.g.
ACS Appl. Mater. Interfaces 2022, https://doi.org/10.1021/acsami.1c17283
- In the introduction section, the authors should highlight the main properties of MXenes and their utilization in various applications and should cite more recent references about MXenes like
Separation and Purification Technology, 2022; Mater. Chem. A, 2022; Electroanalysis 2021
- The authors can write more references about MXenes from Nanomaterials journals like Nanomaterials.
- In the conclusion section, the authors should summarize the most important results and add some values
Author Response
Response to Reviewer 2 Comments
Dear Editors and Reviewers:
Thank you very much for your letter and for the reviewers’ comments on our manuscript entitled “Theoretical Study on Electronic Structure and Magnetic Properties Regulation of Janus Structure of M'MCO2 2D MXenes” (ID: 1569494). Those comments are valuable and very helpful for revising and improving our manuscript. We have considered these comments carefully and revised our manuscript accordingly. Changes are marked in red in the revised manuscript. In the following, we address the reviewers’ comments point by point.
Point 1: The figures numbers should be revised; Figure 4 is missing
Response 1: Thanks for your carefully review. I have midified in the manuscript.
Point 2: The authors should improve the resolution of the whole figures in the manuscript especially Figure 3-Figure 7
Response 2: Thank you for your suggestion. I have change figures in the manuscript.
Point 3: In section 2 ‘’Materials and Methods’’ the authors claimed that ‘’ A vacuum spacing of 20 Å along the M'MCO2 normal was used to avoid ‘’ This spacing is very large compared to that usually observed in the experimentally prepared MXenes the authors should explain.
Response 3: Thanks for asking this question. The calculations using VASP software is based on periodic boundary conditions, so the actual calculation model in c direction is an infinite repetition model composed of atomic layer/vacuum layer/atomic layer/vacuum layer /... . Therefore, when the vacuum layer is relatively thin, the atomic layer will interact with its mirror image structure, resulting in errors. However, if the vacuum layer is too thick, the calculation will increase. Generally, vacuum spacing of 20 Å can avoid the interaction caused by the periodic boundary condition.
Point 4: Also, the authors should explain why he selected this formula M'MCO2 MXenes, sually, Mxenes always have surface terminations of F, OH, and O.
Response 4: Thanks for asking this question. Because O needs more electron than F and OH, and O termination binds more stable to bare M'MC.
Point 5: The authors should add some values and numbers into the abstract
Response 5: Thanks for your suggestion. I have added it in the manuscript.
Point 6: The authors should compare the obtained results with elsewhere reports
Response 6: Thanks for your suggestion. I have compared previous reports with my results in the manuscript.
Point 7: The similarity rate is around 23%, according to the iThenticate, so the authors should decrease it to be lower than 20 %
Response 7: Thanks for your carefully review. I have made some corrections in the manucript.
Point 8: In the introduction section, the authors should extend the introductory section about 2D materials and emphasize their advances over other materials, and can cite more references. e.g. ACS Appl. Mater. Interfaces 2022, https://doi.org/10.1021/acsami.1c17283
Response 8: Thank you for your advice. I have extend the introductory section about 2D materials and cite more references in the manuscript.
Point 9: In the introduction section, the authors should highlight the main properties of MXenes and their utilization in various applications and should cite more recent references about MXenes like Separation and Purification Technology, 2022; Mater. Chem. A, 2022; Electroanalysis 2021
Response 9: Thank you for your advice. I have cited them in the manuscript.
Point 10: The authors can write more references about MXenes from Nanomaterials journals like Nanomaterials.
Response 10: Thank you for your advice. I have cited them in the manuscript.
Point 11: In the conclusion section, the authors should summarize the most important results and add some values
Response 11: Thank you for your advice. I have midified in the manuscript.

Reviewer 3 Report
The authors present a theoretical study of M'MCO2 MXenes. M' and M are V,Cr, and Mn, but M' and M differ.
They expore the electronic structure and magnetic properties using DFT calculations.
The paper probably deserve to be published. But some points must be clarified before the acceptation.
- I do not well understand the sentence "The specific value U does not change the predicted magnetic ordering nor the easy axis determination." I suppose U=0 or U=4eV do not give the same result. Which values are used in the present work ?
-Page 4, line 121. The use of the terminology "non-polarized" vs "polarized" system could be confuse. Please explain what calculations are made in both cases.
- page 4, line 126. I do not understand the statement: "V2CO2 is nonmagnetic (NM). Actually, Magnetic and NM states are degenerate (Table 2).
- Are the geometries of NM, FM, AFM states different ?
- part 3.2 Magnetic moment. Please explain why U is not added in the case of NM state ? I do not understand what calculations are made.
- Figure 3 shows the magnetic state configurations. I am wondering how select one given configuration in a DFT calculation.
- the energy cutoff of 600 eV (said two times in Part 2)
Author Response
Dear Editors and Reviewers:
Thank you very much for your letter and for the reviewers’ comments on our manuscript entitled “Theoretical Study on Electronic Structure and Magnetic Properties Regulation of Janus Structure of M'MCO2 2D MXenes” (ID: 1569494). Those comments are valuable and very helpful for revising and improving our manuscript. We have considered these comments carefully and revised our manuscript accordingly. Changes are marked in red in the revised manuscript. In the following, we address the reviewers’ comments point by point.
Point 1: I do not well understand the sentence "The specific value U does not change the predicted magnetic ordering nor the easy axis determination." I suppose U=0 or U=4eV do not give the same result. Which values are used in the present work ?
Response 1: Thank you for asking the question. I selected U=4eV in the present work. U is the potential of electrons interacting with opposite spins in the same orbital. The correction of DFT+U can consider the in-position Coulomb interaction between electrons, better describe the d orbital electrons with strong correlation, and describe the magnetic moment more accurately, but at the same time, the energy term introduced will increase the total energy of the system.
Point 2: Page 4, line 121. The use of the terminology "non-polarized" vs "polarized" system could be confuse. Please explain what calculations are made in both cases.
Response 2: Thank you for asking the question. There are no “non-polarized" and "polarized" terminology in my manuscript, and I used non-spin polarized and spin-polarized. Non-spin polarized applied in the calculations of nonmagnetic system, while spin polarized applied in the calculations of magnetic system.
Point 3: page 4, line 126. I do not understand the statement: "V2CO2 is nonmagnetic (NM). Actually, Magnetic and NM states are degenerate (Table 2).
Response 3: Thank you for asking this questions. For V2CO2 structure, the total energy of NM state and Magnetic state are -45.5975 and -45.5973 eV/ u.c (Unit cell) respectively. The total energy of NM state is slightly less than Magnetic state. Therefore, the ground state of V2CO2 is nonmagnetic (NM).
Point 4: Are the geometries of NM, FM, AFM states different ?
Response 4: Thank you for asking this questions. For the same M'MCO2, the geometries of NM, FM, AFM state is same. But different state set different initial magnetic moment direction.
Point 5: part 3.2 Magnetic moment. Please explain why U is not added in the case of NM state ? I do not understand what calculations are made.
Response 5: Thank you for asking this questions. U is added to better describe the strongly correlated d orbital electrons, and improve the prediction of magnetic moment and bandgap of the transition metal. So it is meaningless to add U in NM state.
Point 6: Figure 3 shows the magnetic state configurations. I am wondering how select one given configuration in a DFT calculation.
Response 6: Thank you for asking this question. First, determine if your configuration contains magnetic atoms. If no magnetic atoms in your configurations, you should select NM state. Otherwise we should calculate the energy of different magnetic state. The lowest energy state is the magnetic state.
Point 7: the energy cutoff of 600 eV (said two times in Part 2)
Response 7: Thank you for the tip. I have midified in the manuscript.

Reviewer 4 Report
The paper presents the results of theoretical studies of MXenes. The results are interesting and look reliable. The presentation is clear. Hence I think that the paper can be published in the Journal.
Author Response
Dear Editors and Reviewers:
Thank you very much for your letter and for the reviewers’ comments on our manuscript entitled “Theoretical Study on Electronic Structure and Magnetic Properties Regulation of Janus Structure of M'MCO2 2D MXenes” (ID: 1569494).
With best wishes!
Ms. Ping Qian
